# Anti-Obesity and Anti-Adipogenic Effects of Administration of Arginyl-Fructose-Enriched Jeju Barley (*Hordeum vulgare* L.) Extract in C57BL/6 Mice and in 3T3-L1 Preadipocytes Models

**DOI:** 10.3390/molecules27103248

**Published:** 2022-05-19

**Authors:** Soo-Young Lee, Tae-Yang Kim, Ji-Yoon Hong, Gi-Jung Kim, Jung-Bae Oh, Min-Joo Kim, Emmanouil Apostolidis, Jung-Yun Lee, Young-In Kwon

**Affiliations:** 1Department of Food and Nutrition, Hannam University, Daejeon 34054, Korea; dltndud1221@naver.com (S.-Y.L.); xodid5606@naver.com (T.-Y.K.); ysmieee@naver.com (J.-Y.H.); homina97@daum.net (G.-J.K.); minjookim@hnu.kr (M.-J.K.); 2Institute of Functional Foods, Kunpoong Bio Co., Ltd., Jeju 63010, Korea; denisoh89@gmail.com; 3Department of Chemistry and Food Science, Framingham State University, Framingham, MA 01701, USA; eapostolidis@framingham.edu

**Keywords:** Amadori rearrangement compounds, barley, anti-obesity, arginyl-fructose, adipogenic

## Abstract

In our previous study, we reported that arginyl-fructose (AF), one of the Amadori rearrangement compounds (ARCs) produced by the heat processing of Korean ginseng can reduce carbohydrate absorption by inhibiting intestinal carbohydrate hydrolyzing enzymes in both in vitro and in vivo animal models. This reduced absorption of carbohydrate might be helpful to control body weight gain due to excessive carbohydrate consumption and support induced calorie restriction. However, the weight management effect, except for the effect due to anti-hyperglycemic action, along with the potential mechanism of action have not yet been determined. Therefore, the efforts of this study are to investigate and understand the possible weight management effect and mechanism action of AF-enriched barley extracts (BEE). More specifically, the effect of BEE on lipid accumulation and adipogenic gene expression, body weight gain, body weight, plasma lipids, body fat mass, and lipid deposition were evaluated using C57BL/6 mice and 3T3-L1 preadipocytes models. The formation of lipid droplets in the 3T3-L1 treated with BEE (500 and 750 µg/mL) was significantly blocked (*p* < 0.05 and *p* < 0.01, respectively). Male C57BL/6 mice were fed a high-fat diet (30% fat) for 8 weeks with BEE (0.3 g/kg-body weight). Compared to the high fat diet control (HFD) group, the cells treated with BEE significantly decreased in intracellular lipid accumulation with concomitant decreases in the expression of key transcription factors, peroxisome proliferator-activated receptor gamma (PPARγ), CCAAT/enhancer-binding protein alpha (CEBP/α), the mRNA expression of downstream lipogenic target genes such as fatty acid binding protein 4 (FABP4), fatty acid synthase (FAS), and sterol regulatory element-binding protein 1c (SREBP-1c). Supplementation of BEE effectively lowered the body weight gain, visceral fat accumulation, and plasma lipid concentrations. Compared to the HFD group, BEE significantly suppressed body weight gain (16.06 ± 2.44 g vs. 9.40 ± 1.39 g, *p* < 0.01) and increased serum adiponectin levels, significantly, 1.6-folder higher than the control group. These results indicate that AF-enriched barley extracts may prevent diet-induced weight gain and the anti-obesity effect is mediated in part by inhibiting adipogenesis and increasing adiponectin level.

## 1. Introduction

The energy imbalance resulting from excessive food intake and increased caloric intake results in obesity, a serious health concern that can lead to increased incidence of chronic diseases such as coronary heart disease, diabetes, nonalcoholic fatty liver disease (NAFLD), and cancer [1]. Obesity can be characterized by weight gain resulting from excessive body fat accumulation. Genetic and environmental factors may induce weight gain, but it has been clearly defined that calorie intake is a predominant factor. Diet-induced obesity (DIO) animal models are commonly used to study obesity since these models reflect the state of human obesity much better than genetically modified models [2]. These models are based on the fact that a high fat diet increased weight gain and initiated the diabetic stage in mice and rats [3,4].

Adipogenesis is a process in which preadipocytes are hyperplastically transformed into adipocytes, resulting in hypertrophy and dysfunctional adipocytes due to the excessive storage of lipids as triglycerides through increased adipogenesis. It has been reported that adipogenesis and adipogenesis are initiated by the expression of differentiation-related transcription factors when preadipocytes are exposed to adipogenic inducers. Recent evidence suggests that dietary factors containing several bioactive compounds can effectively change adipocyte number as well as adipocyte size by regulating the expression of these adipocyte differentiation-related transcription factors [5,6], potentially resulting in fewer adipocytes. Therefore, the regulation of key transcription factors such as PPARγ and C/EBPα in the early stage of adipogenesis, and the control of expression of fatty acid synthase (FAS), lipoprotein lipase (LPL), fatty acid binding protein 4 (FABP), and hormone sensitive lipase (HSL) are being studied in various strategies using bioactive compounds derived from foods, especially in plant foods.

Barley is a widely consumed cereal among the most ancient cereal crops. Almost 80–90% of barley production is for animal feeds and malt, but now barley is gaining renewed interest as an ingredient for the production of functional foods due to their concentration of bioactive compounds, such as β-glucans and tocols [7]. Moreover, there are many classes of phenolic compounds in barley, such as benzoic and cinnamic acid derivatives, proanthocyanidins, quinines, flavonols, chalcones, flavones, flavanones, and amino phenolic compounds [8]. Barley is traditionally roasted in some Asian countries to produce a beverage (Japanese roasted barley tea, Mugicha) and during this roasting process various aroma compounds are formed [9]. The predominant amino acids present in barley have been reported to be glutamine and proline, but arginine is also found in significant levels [10], and barley is also an excellent source of glucose due to the presence of maltose.

During the heat process of food products that contain reducing sugars and amino acids, non-enzymatic browning (or Maillard reaction) takes place [11]. More specifically, food products containing arginine and glucose or maltose and undergoing heat processing, initially form arginyl-fructose (AF) and arginyl-fructosyl-glucose (AFG) through Amadori rearrangement (ARC) in products such as grains [12]. Previous reports have demonstrated that AF and AFG can modulate glucose uptake from dietary carbohydrates by exhibiting an inhibitory effect on α-glucosidases [12,13]. Furthermore, AF’s anti-hyperglycemic effect has been demonstrated by using Korean Ginseng as a model food, in both animal and clinical models [14,15]. Through this mechanism, the inhibiting carbohydrate hydrolyzing enzymes, ARCs, additionally contribute towards the reduction in the caloric uptake, which can in turn result in weight and fat gain management.

The development of melanoidins and their antioxidant effects in roasted barley have been investigated in the past [16]. However, very limited information is available about the development of the ARC’s content in roasted barley, other than aroma compounds, and their subsequent enhanced anti-obesity effect. Therefore, to evaluate the effect of ARC-rich barley extract on weight management and investigate the possible mechanism of action on improving weight gain, it is essential to measure the changes in body weight gain, body weight, plasma lipids, body fat mass, and lipid deposition. Consequently, the purpose of this study was (i) to investigate the changes in physical condition and (ii) fat metabolism-related biochemical markers such as transcriptional factors, gene expression, and (iii) to measure the changes in cell-based fat accumulation and its morphology on supplementation of heat-processed/AF-enriched barley extract (BEE) in C57BL/6 mice fed high-fat diets.

## 2. Results

### 2.1. BEE Inhibits Adipocyte Differentiation

Initially we evaluated the possible cytotoxicty of BEE on preadipocyte 3T3-L1 cells. As shown in Figure 1a, when 3T3-L1 cells were treated with 250 to 750 µg/mL of BEE for 48 h at 37 °C, our findings demonstrated no cytotoxicity at the tested doses (Figure 1a). Adipogenesis during the differentiation of 3T3-L1 preadipocytes into mature adipocytes was evaluated with the supplementation of BEE. These results suggest that higher concentrations of BEE administration (250 to 750 µg/mL) inhibit adipogenesis (Figure 1b). More specifically, we observed that BEE inhibited lipid accumulations in a dose-dependent manner. The formation of lipid droplets in the 3T3-L1 treated with BEE (500 and 750 µg/mL) was significantly blocked (*p* < 0.05 and *p* < 0.01, respectively, Figure 1b).

### 2.2. BEE Alleviates HFD-Induced Obesity in In Vivo Model

To prove the results from the in vitro experiment, the effects of BEE administration (0.3 g/kg-body weight) were evaluated in a high fat diet (HFD)-induced obesity C57BL/6 mice model for 54 days with dietary composition described in the section of material and methods. BEE supplementation, along with an HFD, resulted in significant changes in food intake and weight gain after day 24, compared to the control group (Figure 2). More specifically, by the last day of the experiment, there was no significant difference in food consumption between the control and BEE group (Figure 2a). Additionally, BEE consumption significantly reduced weight gain compared to the control (Figure 2b).

By looking at specific parameters, we observed that although the initial weight of the mice in the control and BEE groups were similar, a significant reduction was observed at the end of the experiment in the BEE-treated animals when compared to the control (HFD) (*p* < 0.01, Table 1). By estimating the changes from initial weight to final weight, BEE supplementation reduced weight gain by around 35% compared to the control.

Interestingly, after the 24th day, in the case of the BEE-administered group, food intake increased, but compared to the HFD control group, the body weight decreased. These results are considered to be related to the postprandial blood glucose rise inhibitory effect of the arginyl-fructose (main key compound in BEE) reported in the previous study [12,13]. In addition to the regulation effect of BEE on adipogenesis/lipogenesis, it indicates that the effect of inhibiting the absorption of ingested carbohydrates into the small intestine also may act, giving a synergistic effect.

Additionally, we observed that BEE administration significantly reduced the LDL cholesterol (*p* < 0.01) and total triglyceride (*p* < 0.01) levels, and increased HDL cholesterol (*p* < 0.05) when compared to the control (HFD) (Table 1, Figure 3). As shown in Table 2, treatment with BEE also markedly decreased the weight of fat mass for each fat tissue location (Table 2).

Additionally, BEE administration resulted in a significant increase in the levels of the hormone adiponectin (*p* < 0.01) (relevant to insulin sensitivity), TNF-α (*p* < 0.05), while reducing leptin (*p* < 0.01) (suggesting improvement of leptin resistance) and insulin levels (*p* < 0.01) when compared to the control (HFD) (Table 3).

### 2.3. BEE Administration Decreases the Expression of Adipogenesis/Lipogenesis-Related Genes in Epididymal Fat Biopsy from C57BL/6 Mice

CEBPα, FAS, PPARγ, and SREBP-1c are the main key genes of fat metabolism and are involved in the regulation of fat metabolism. Increased CEBPα expression is related to adipocyte hypertrophy, impaired insulin signaling, and decreased glucose utilization [17]. It has been well reported that decreased FAS and PPARγ expressions are considered as regulation factors of adipogenesis [18]. SREBP-1c participated in adipocyte differentiation and adipogenesis is a major regulator of lipid homeostasis transcription [19]. Our observations demonstrated that PPARγ (*p* < 0.05) and C/EBPα (*p* < 0.001) mRNA levels were significantly decreased with the administration of BEE (0.3 g/kg-body weight) (Figure 4b,d). Additionally, we observed that BEE supplementation markedly decreased the expression levels of key target lipogenic genes, fatty acid synthase (FAS) (*p* < 0.001), lipoprotein lipase (LPL) (*p* < 0.01), and fatty acid binding protein 4 (FABP4) (*p* < 0.001), and sterol regulatory element-binding protein 1c (SREBP-1c) (*p* < 0.01) (Figure 4a,c,e,f), an observation that correlates with the reduced PPARγ and C/EBPα mRNA expression. The above lipid metabolism results indicated that BEE treatment affected the gene and/or protein expression of CEBPα, FAS, SREBP-1c, and PPARγ in C/57BL6 related to lipid metabolism.

### 2.4. BEE Administration Decreases the Fat Accumulation in the HFD-Induced C57BL/6 Mice

Our initial study with preadipocyte 3T3-L1 examined whether BEE treatment markedly alleviates hepatic fat accumulation in HFD-induced C57BL/6 mice. We investigated the liver weight of mice after 8 weeks of BEE administration. While the liver weight was found to be increased in HFD mice, the BEE treatment prevented this increase in the HFD mice (Figure 5b,c).

Diet-induced obesity through the administration of a western-type diet high in fat and sugar to mice can result in a higher weight of adipose and liver tissues. The increased adipose tissue weight can be due to hypertrophy (bigger cells) or hyperplasia (more cells) of adipocytes, or a combination of both. Hematoxylin and eosin (H&E) staining of adipose tissue sections was conducted to compare the sizes of the individual adipocytes.

For the BEE treatment group, the adipose tissue mass (subcutaneous and epididymal fat) was significantly lower than the HFD group (Table 2). The difference in adipocyte size was evident in H&E-stained samples of both normal control and HFD. Moreover, the BEE treatment group’s adipose tissues showed smaller adipocytes than the HFD adipose tissue (Figure 6b,c).

## 3. Discussion

In this paper we present the anti-obesity effects of BEE in a rodent preadipocyte cell line, 3T3-L1 cells, and also in a C57BL/6 mice model of HFD-induced obesity. Our findings suggest the dose-dependent inhibition of lipid accumulation and 3T3-L1 preadipocyte differentiation with BEE supplementation. Additionally, BEE decreased the expression of PPARγ and C/EBPα, which are important transcription factors regulating adipocyte metabolism, and also decreased the expression of their downstream target genes including FABP4, LPL, and FAS. The latter genes are adipocyte-specific and are involved in maintaining the adipocyte phenotype. Additionally, we confirmed that the tested doses of BEE do not have any toxicity against 3T3-L1 cells.

The anti-obesity effect of BEE administration was evaluated in a mice model for 42 days and we observed that BEE supplementation decreased body weight compared to the control without a significant difference in food intake. In addition, BEE administration reduced the levels of triglycerides, total cholesterol, and LDL cholesterol levels while increasing the HDL cholesterol level.

Dietary fat intake and obesity have been strongly correlated through numerous epidemiological studies [20]. Hypertrophic adipocytes and the dysfunction of adipose tissue are the main obesity characteristics. Reducing the uptake of dietary lipids in adipocytes can result in elevated triglycerides and reduced HDL cholesterol levels. It has been well-defined that adipocyte differentiation is characterized by the increased expression of PPARγ and C/EBPα [21]. We observed that BEE treatment significantly reduced levels of PPARγ and C/EBPα. We also observed a significant reduction in FABP4, FAS, and LPL gene expression. The correlation between PPARγ and C/EBPα and lipogenesis has been well-defined, but our findings further confirm it. Our findings in the animal trial suggest that BEE can result in reduced body weight, triglyceride, and total cholesterol while increasing HDL cholesterol. Based on these observations, we suggest that BEE can possibly be used as a weight management strategy through adipogenesis inhibition.

Adiponectin is a hormone secreted by adipocytes that plays an important role in maintaining lipid homeostasis. Adiponectin is secreted from adipose tissue and binds to adiponectin receptors (adipoR1 and adipoR2) in the liver, enhancing insulin sensitivity while maintaining low levels of lipids in the liver through controlling lipogenesis, and increasing β-oxidation through the activation of AMP protein kinase. Obesity downregulates adiponectin secretion, leading to excessive lipid accumulation including triglyceride and LDL cholesterol. As a result, higher levels of adiponectin are important in the prevention of obesity. Our observations suggest that BEE administration increases the adiponectin levels, which can be an additional mechanism for the weight gain reduction effect observed in this study.

Leptin is a hormone produced by the adipose tissue and communicates the state of body energy repletion to the central nervous system (CNS) in order to suppress food intake and permit energy expenditure [22,23,24]. Most obese individuals exhibit elevated circulating leptin levels commensurate with their adipose mass, a condition defined as leptin resistance [25]. In our study we observed that leptin levels were higher in the HFD group when compared to the BEE group. This suggests that leptin resistance might have developed in the HFD group and not in the BEE-treated group, indicating that BEE prevents the development of leptin resistance. This could be another mechanism for the observed weight management effects of BEE supplementation.

In conclusion, BEE administration reduces weight gain and body fat accumulation in a mouse model. Our observations provide strong supporting evidence for additional studies to better define the potential weight management effect of BEE. Here we report a mechanism through the management of adipocyte differentiation and adiponectin secretion. Our previous research results related to arginyl-fructose, a major active ingredient in BEE, were that AF showed a calorie-restriction effect [12,13]. Considering this, AF, the main component of BEE, also inhibited the activity of α-glucosidase in the small intestine and blocked the absorption of disaccharides, which may have a synergistic effect on the anti-adipogenesis/obesity efficacy, resulting in a statistically significant decrease in weight gain. In the future, it is important to determine how BEE might affect glucose and lipid metabolism in various relevant tissues, such as skeletal muscle and the liver.

## 4. Materials and Methods

### 4.1. Materials

Arginyl-fructose (AF)-enriched barley powder (BEE) was purchased from Kunpoong Bio Co., Ltd. (Jeju, Korea). Corn starch, casein, vitamin mix, mineral mix, calcium phosphate, and sodium chloride were purchased from Raon Bio (Yonginsi, Korea). Total cholesterol and total glyceride kits were purchased from Stanbio laboratory (Boerne, TX, USA). Unless noted, all chemicals were purchased from Sigma-Aldrich Co. (St. Louis, MO, USA). The fast SYBR real-time PCR master mix, Dulbecco’s modified Eagle’s medium (DMEM), fetal bovine serum (FBS), bovine calf newborn serum (BCS), penicillin-streptomycin (*p*/S), and trypsin-EDTA were obtained from Life Technologies (Grand Island, NY, USA). Adiponectin ELISA kit was purchased from Thermo Fisher Scientific (Invitrogen, Carlsbad, CA, USA). 3T3-L1 cells (ATCCV^®^CL-173TM) were used below passage 12. 3T3-L1 preadipocytes were propagated and cultured in DMEM medium supplemented with 10% BCS and 1% P/S until confluent and maintained for additional 2 days and differentiated as reported previously [26] with or without BEE.

### 4.2. Sample Preparation

In order to obtain an arginyl-fructose (AF)-enriched barley powder (BEE) sample, gelatinization was performed at 80 °C for 1 h in 500 L distilled water after milling and pulverizing of 25 kg of barley. Then, 50 mL of a triple complex enzyme (AMG^®^ glucoamlyase, Viscoflow^®^ MG, Pectinex^®^ (1:1:1 ratio), Bagsvaerd, Denmark) and 5.9 kg of citric acid were added and reaction was performed at 60 °C for 18 h. Arginine (14.9 kg) was added to adjust the neutral pH. To maximize the content of the Amadori derivative, heating was carried out at 95 °C for 4 h. Filtration of mixture was carried out using a filter paper, and 54.2 kg of dextrin was added. To obtain Amadori derivative containing barley powder, spray drying followed at 170 °C. AF content in BEE was 22%, approximately.

### 4.3. Determination of Cell Viability and Morphometric Analysis

The effects of BEE on 3T3-L1 cell viability were determined using an established MTT assay. Briefly, the 3T3-L1 preadipocytes cells were seeded at a density of 1 × 104 cells per well in a 96-well plate and incubated in culture medium at 37 °C for 24 h to allow at-tachment. The attached cells were either untreated control or treated with 250, 500, or 750 µg/mL of BEE at 37 °C for 48 h. After 48 h of incubation, the cells were washed with phosphate-buffered saline (PBS) prior to the addition of MTT (0.5 µg/mL PBS) and incubated at 37 °C for 2 h. Formazan crystals were dissolved with dimethyl sulfoxide (100 µL/well) and detected at OD570 with a model Emax (Molecular Devices, Sunnyvale, CA, USA).

Fresh dissected mice epididymal adipose and liver tissue samples (*n* = 5) were fixed by immersion in 4% paraformaldehyde in 0.1 M phosphate buffer (pH 7.4) overnight at 4 °C. Samples were then washed in phosphate buffer and dehydrated in a graded series of ethanol, cleared in xylene, and embedded in paraffin blocks. Five-micrometer-thick sections of the tissues were stained with hematoxylin and eosin to assess morphology. Images from light microscopy were digitalized and the diameters of at least 100 adipocytes of each section were determined using Axio Vision software (Carl Zeiss Imaging Solutions).

### 4.4. Oil Red O (ORO) Staining

To determine the degree of differentiation as measured by intracellular lipid content, ORO was performed as previously described [27]. Briefly, 3T3-L1 preadipocytes were cul-tured in DMEM/high-glucose medium containing 10% calf serum until confluent (day −2) and maintained for an additional 2 days (until day 0). Differentiation was induced on day 0 by the addition of 0.5 mmol/L methylisobutylxanthine, 1 μmol/L dexamethasone, 1.0 μg/mL insulin, and 10% fetal bovine serum (FBS) in DMEM. After 48 h (day 2), the medium was replaced with DMEM containing 1.0 μg/mL insulin and 10% FBS. Medium was changed every 2 days thereafter until the cells were collected for analysis [18]. BEE was reconstituted as 1000 µg/mL stock solutions in DMSO (dimethyl sulfoxide) and added at the indicated concentrations on day 0. Cells were cultured with BEE until cells were collected for analysis. After 8 days of differentiation, 3T3-L1 adipocytes were washed with 4% paraformaldehyde once and fixed with 4% paraformaldehyde for 20 min at room temperature. Cells were then washed with 60% isopropanol once and stained with diluted Oil Red O solution for 30 min. After photographing the stained cells, the dye retained in 3T3-L1 cells was eluted with 100% isopropanol and the absorbance was measured by a microplate reader (SpectraMax M2, Molecular Devices, Sunnyvale, CA, USA) at 490 nm.

### 4.5. Quantitative Real-Time PCR

RNA was isolated with TRIzol^®^ plus RNA purification kit according to manufacturer’s protocol (Life Technologies, Grand Island, NY, USA). One microgram of total RNA was used to synthesize cDNA using Revert Aid First Strand cDNA Synthesis kit (Thermo Scientific, Waltham, MA, USA). The reaction was performed with Fast SYBR^®^ Green Master Mix containing 1 µM of primer pair and 100 ng of cDNA under 40 cycles with each of 95 °C for 1 sec and 58 °C for 20 s. Relative levels of the target mRNA expression were determined by ViiA^TM^ 7 real-time PCR system (Life technologies, Grand Island, NY, USA), normalized to GAPDH calculated with the 2−(^ΔΔ^nd) method. The primer sequences are listed in Table 4. All the results were normalized to the housekeeping gene, glyceraldehyde 3-phosphate dehydrogenase (GAPDH), to control for variations in mRNA concentrations. Relative quantification was performed using the comparative delta-delta Ct method according to the manufacturer’s instructions (Applied Biosystems).

### 4.6. In Vivo Experimental Design

Five-week-old male C57BL/6 mice were purchased from Joongang Experimental Animal Co. (Seoul, Korea) and fed a high fat diet (30% fat) (Table 5) for 54 days. After 3 days the normal diet (Pico 5053) was switched to a high fat diet (HFD) (Oriental Bio. Co., Seongnam, Korea) for 54 days. During the HFD administration for 8 weeks, C57BL/6 mice were divided into two groups, one group received HFD and the second group received HFD with BEE. BEE was orally administrated 2 times a day (9–10 a.m. and 4–5 p.m.). Distilled water (D.W.) was used as a vehicle for a solution of the experimental compound, BEE. In a control group, D.W. was used as a vehicle for oral administration without BEE using a zonde injection needle. The dose of each BEE administration was 0.15 g/kg-body weight, yielding a final dose of 0.3 g/kg-body weight/day. The animals were housed in individual cages in a room with a 12 h light/dark cycle (lights on from 06:00 h) with 50% ± 7% relative humidity. In this study, ten C57BL/6 mice were used for each group. The experimental protocols were approved by the Institutional Animal Care and Use Committee (IACUC) of the Hannam University (Approval number: HNU2020-0014). The mice had free access to water throughout the experimental period. The mice were anesthetized with pentobarbital and sacrificed, and blood was collected and serum was processed and stored at −80 °C until used. The retroperitoneal, mesenteric, kidney, subcutaneous, and epididymal fat tissue depots were rapidly removed and weighed. Samples for RNA and protein analysis, and for DNA quantification were immediately frozen in liquid nitrogen and stored at −70 °C until analysis.

Group I: Control.

Group II: BEE 0.3 g/kg-body weight/day.

### 4.7. Blood Analysis

The plasma total cholesterol and total glyceride concentration was measured using a kit (Stanbio lab., Boerne, TX, USA). Serum adiponectin levels in SD rats were detected by ELISA kit (Invitrogen, Carlsbad, CA, USA).

### 4.8. Statistical Analysis

Statistical analyses were carried out using the statistical package SPSS 10 (Statistical Package for Social Science, SPSS Inc., Chicago, IL, USA) program and significance of each group was verified with the analysis of One-way analysis of variance (ANOVA) followed by the Duncan’s multiple range test of *p* < 0.05 and the Student’s *t*-test for comparison of means.

## Figures and Tables

**Figure 1 molecules-27-03248-f001:**
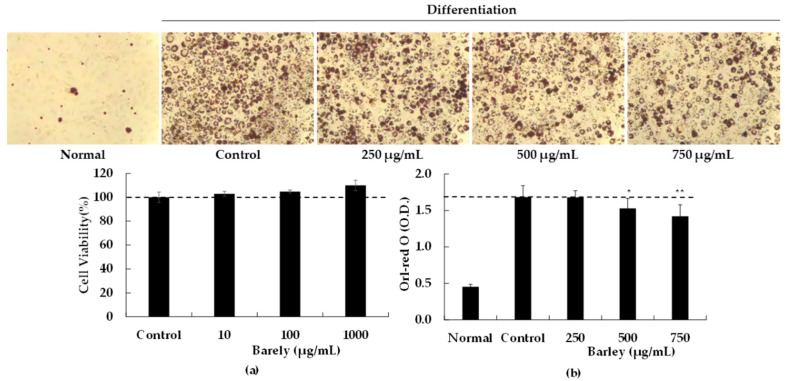
Effects of BEE on the lipid accumulation of 3T3-L1 cells. (**a**): 3T3-L1 preadipocytes cells were seeded at a density of 1 × 104 cells per well in 96-well plates and incubated in culture medium at 37 °C for 24 h to allow attachment. The attached cells were either untreated control or treated with 10, 100, or 1000 µg/mL of BEE at 37 °C for 48 h. After 48 h of incubation. The effects of BEE on cell viability were measured by MTT assay. The data are presented as relative cell viability values. Data are the means ± standard deviation (S.D.) values of at least 3 independent experiments. (**b**): 3T3-L1 preadipocytes were grown and differentiated with the differentiation cocktail in the absence and presence of varying concentrations (0, 250, 500, and 750 μg/mL) of BEE throughout the differentiation for 8 days. After 8 days of differentiation, these cells were subjected to Oil Red O staining for control and BEE to compare intracellular lipid accumulation (control: without BEE, normal: no differentiation). The results are expressed as the mean ± S.D (*n* ≥ 6). Significantly different from control group (* *p* < 0.05, ** *p* < 0.01).

**Figure 2 molecules-27-03248-f002:**
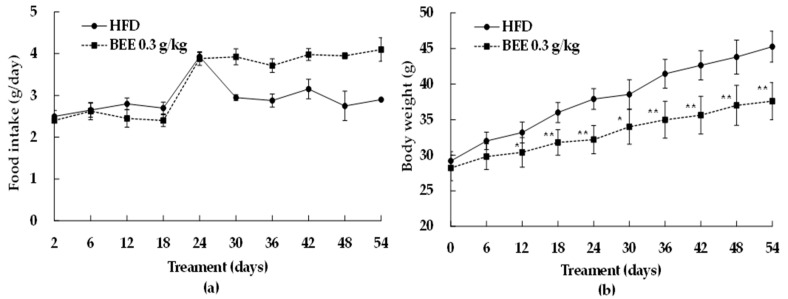
Changes in food intake (**a**) and body weight gains (**b**) before/after administration of BEE. Male C57BL/6 mice had free access to a high fat diet (HFD) (30% fat) to induce the weight gain for 54 days. During an HFD administration for 54 days, BEE was orally administrated (0.3 g/kg-body weight/day, peroral zonde injection) to BEE group C57BL/6 mice, 2 times per day (9–10 a.m. and 4–5 p.m.) with 0.15 g/kg-body weight each. Ten C57BL/6 mice were used for each group. Each point represents mean ± S.D. (*n* = 10). Food intake and body weight levels were compared between control (HFD) and treatment group (BEE) at each time point by unpaired Student’s *t*-test (* *p* < 0.05; ** *p* < 0.01).

**Figure 3 molecules-27-03248-f003:**
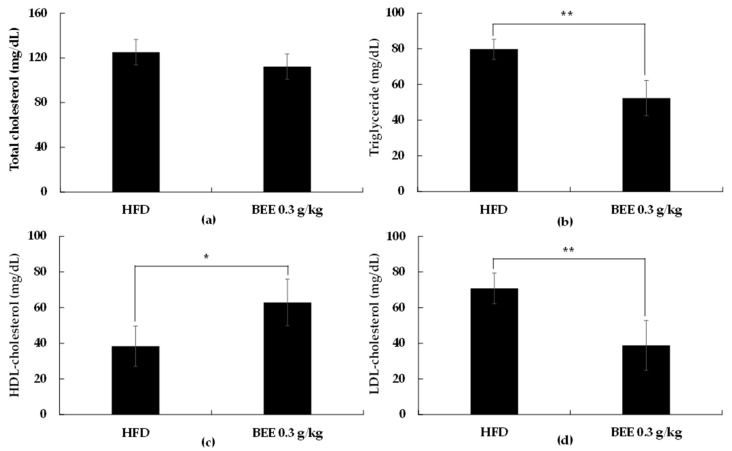
Comparison of total cholesterol (**a**), triglyceride (**b**), HDL (**c**), and LDL (**d**) contents with or without administration of BEE. Each point represents mean ± S.D. (*n* = 10). Triglyceride, total cholesterol, HDL, LDL contents were compared between control (HFD) and treatment group (BEE) at each time point by unpaired Student’s *t*-test (* *p* < 0.05; ** *p* < 0.01).

**Figure 4 molecules-27-03248-f004:**
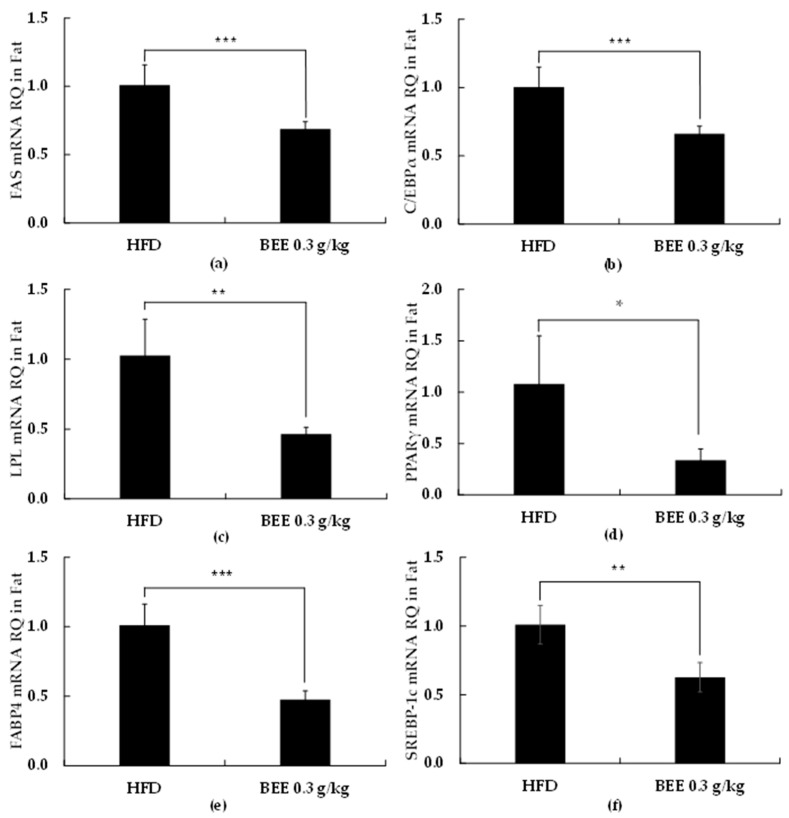
RT, real-time PCR quantitative analysis of adipocyte differentiation-related genes expression in the 3T3-L1. For real-time PCR, we used SYBR green mix with gene-specific primers ((**a**): fatty acid synthase (FAS), (**b**): CEBP/α, (**c**): lipoprotein lipase (LPL), (**d**): PPARγ, (**e**): FABP4, and (**f**): SREBP-1c). Each value is expressed as mean ± S.D. and is representative of at least three separate experiments. Different letters indicate statistically significant differences between groups with one-way ANOVA followed by Duncan’s test of *p* < 0.05. The results are expressed as the mean ± S.D (*n* ≥ 6). Statistical significances from control group were determined by Student’s *t*-test (* *p* < 0.05, ** *p* < 0.01, *** *p* < 0.001).

**Figure 5 molecules-27-03248-f005:**
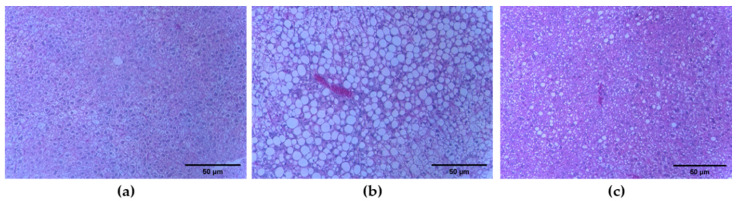
Effect of Jeju barley extract (BEE) on the histopathological change in liver tissues in the high fat diet-induced C57BL/6 mice. Liver tissues were stained with H&E (original magnification ×100). (**a**) Normal diet mice, (**b**) high fat diet mice, (**c**) high fat diet treated with BEE 0.3 g/kg-body weight.

**Figure 6 molecules-27-03248-f006:**
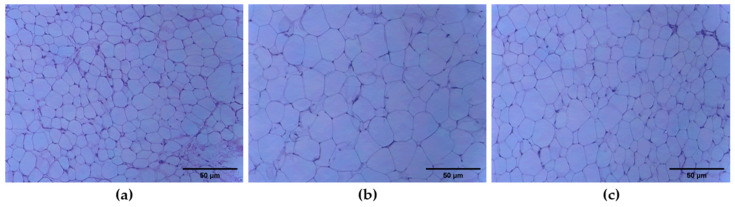
Effect of Jeju barley extract (BEE) on the histopathological change in epididymal fat tissues in the high fat diet-induced C57BL/6 mice. Epididymal fat tissues were stained with H&E (original magnification ×100). (**a**) Normal diet mice, (**b**) high fat diet mice, (**c**) high fat diet treated with BEE 0.3 g/kg-body weight.

**Table 1 molecules-27-03248-t001:** Effects of BEE treatment on various parameters in C57BL/6.

Parameters	C57BL/6	
	HFD	BEE
Initial body weight (g)	28.90 ± 1.66	28.90 ± 1.97
Final body weight (g)	45.26 ± 2.17	37.60 ± 2.60 **
Final body weight gain (g)	16.06 ± 2.44	9.40 ± 1.39 **
FER (%) ^†^	8.70 ± 2.75	4.99 ± 1.63 **
Total cholesterol (mg/dL)	125.10 ± 11.42	112.26 ± 11.35
Triglyceride (mg/dL)	79.76 ± 5.60	52.36 ± 9.81 **
HDL Cholesterol (mg/dL)	38.34 ± 11.28	62.94 ± 13.06 *
LDL Cholesterol (mg/dL)	70.81 ± 8.62	38.85 ± 14.00 **

^†^ FER, Feed efficiency ratio (%) = [Body weight gain (g)/Food intake (g)] × 100. Each experiment was compared between control (HFD) and BEE administration group (BEE) at each time point by unpaired Student’s *t*-test (* *p* < 0.05; ** *p* < 0.01).

**Table 2 molecules-27-03248-t002:** Effects of BEE treatment on various organs and adipose tissue weight in C57BL/6.

Parameters (mg/g)	C57BL/6	
	HFD	BEE
Liver	40.892 ± 4.691	35.059 ± 3.917
Kidney	7.215 ± 0.458	8.321 ± 0.564 **
Cecum	3.433 ± 0.619	4.589 ± 1.084
Mesenteric fat	20.805 ± 1.276	15.337 ± 1.791 **
Retroperitoneal fat	11.945 ± 1.801	11.301 ± 2.289 *
Kidney fat	4.027 ± 2.347	3.177 ± 0.699
Subcutaneous fat	54.673 ± 8.099	33.792 ± 8.907 **
Epididymal fat	49.196 ± 5.720	37.537 ± 6.018 *
Small intestine	17.553 ± 1.853	21.095 ± 4.944

Each experiment was compared between control (HFD) and BEE administration group (BEE) at each time point by unpaired Student’s *t*-test (* *p* < 0.05; ** *p* < 0.01).

**Table 3 molecules-27-03248-t003:** Effects of BEE treatment on various obesity-related hormone and cytokine parameters in C57BL/6.

Parameters	C57BL/6	
	HFD	BEE
Adiponectin (μg/mL)	34.93 ± 8.46	54.81 ± 10.81 *
Leptin (ng/mL)	41.26 ± 8.60	22.19 ± 8.24 **
TNF-α (pg/mL)	31.00 ± 2.58	42.03 ± 9.09 *
IL-6 (μg/mL)	67.55 ± 14.84	41.92 ± 11.82 *
Insulin (μg/mL)	0.77 ± 0.08	0.48 ± 0.06 **

Each experiment was compared between control (HFD) and BEE administration group (BEE) at each time point by unpaired Student’s *t*-test (* *p* < 0.05; ** *p* < 0.01).

**Table 4 molecules-27-03248-t004:** Primer for real-time quantitative PCR.

Genes	Primer Sequences	
Accession Number	Forward (5′-3′)	Reverse (5′-3′)
GAPDH	CGTCCCGTAGACAAAATGGT	TTGATGGCAACAATCTCCAC
NM_008084		
PPARγ	GAAAGACAACGGACAAATCACC	GGGGGTGATATGTTTGAACTTG
NM_011146		
C/EBPα	TTGTTTGGCTTTATCTCGGC	CCAAGAAGTCGGTGGACAAG
NM_007678		
FABP4	AGCCTTTCTCACCTGGAAGA	TTGTGGCAAAGCCCATC
NM_024406		
FAS	TGATGTGGAACACAGCAAGG	GGCTGTGGTGACTCTTAGTGATAA
NM_007988		
SREBP-1c	ACGGAGCCATGGATTGCACA	AAGGGTGCAGGTGTCACCTT
NM_011480		
LPL	GGACGGTAACGGGAATGTATGA	TGACATTGGAGTCAGGTTCTCTCT
NM_008509		

PCR, polymerase chain reaction; PPARγ, peroxisome proliferator-activated receptor γ; C/EBPα, CCAAT/enhancer-binding protein α; FAS, fatty acid synthase; SREBP-1c, sterol regulatory element-binding protein-1c; LPL, lipoprotein lipase; GAPDH, glyceraldehyde 3-phosphate dehydrogenase.

**Table 5 molecules-27-03248-t005:** Composition of high fat diet (g/kg).

High Fat Diets (g/kg)	
Corn starch	321
Sucrose	100
Casein	200
Corn oil	100
Lard	200
Cellulose	30
DL-methionine	2
Vitamin mix ^(^^1)^	10
Mineral mix ^(^^2)^	35
Choline bitartrate	2

^(1)^ Vitamin mixture: AIN-93VX; ^(2)^ Mineral mixture: AIN-93G.

## Data Availability

The data are available on request.

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
