# Peer review of "Anti-Obesity and Anti-Adipogenic Effects of Administration of Arginyl-Fructose-Enriched Jeju Barley (Hordeum vulgare L.) Extract in C57BL/6 Mice and in 3T3-L1 Preadipocytes Models"

_molecules, 2022, doi:10.3390/molecules27103248_

Round 1
Reviewer 1 Report
The authors investigated the anti-obesity effect of Aginyl-fructose (AF)-enriched barley extracts (BEE) on anti-obesity factors such as lipid accumulation, adipogenic gene expression, body weight gain, and plasma lipid levels in high fat diet (HFD)-fed mice and 3T3-L1 adipocytes. BEE suppressed lipid accumulation in 3T3-L1 cells. Administration of BEE to HFD-fed mice decreased body weight gain with the decreased serum cholesterol and triglyceride levels and reduced adipogenic gene expression, compared with vehicle-treated HFD-fed mice. The results are basically sound. However, there are concerns that should be addressed.
- Although in vitro study using 3T3-L1 cells was carried out, the results are not found in Abstract. The description about these results should be included in Abstract.
- The pictures of the cells in Figures 1 and 5 are not clear. They should be exchanged to clearer ones.
- The picture of representative animals after treatment had better been shown.
- Why was body weight gain in the BEE group decreased (Figure 2b), although food intake was increased after 24 days, compared with HFD group (Figure 2a)? The authors should explain this reason.
- Only 54 days-administration is not “long-term”. 54 day-treatment is quite common in these studies. The title should be revised.
- Table 2 and Figure 3 are redundant. Either one should be removed.
- How many animals were used in each study? They should be indicated in the legends of each figure and Table.
- The expression of FAS was lowered in BEE group. However, although the authors stated that FAS is a key factor in lipogenesis, only one gene (FAS) is not enough. To mention the effect to lipogenesis, at least SREBP-1c, a key transcription factor in lipogenesis and other lipogenic gene such as SCD should be analyzed.
Author Response
Thank you for your invaluable advice and review. Corrections and supplements have been completed throughout the entire manuscript for all pointed out points.
The authors investigated the anti-obesity effect of Aginyl-fructose (AF)-enriched barley extracts (BEE) on anti-obesity factors such as lipid accumulation, adipogenic gene expression, body weight gain, and plasma lipid levels in high fat diet (HFD)-fed mice and 3T3-L1 adipocytes. BEE suppressed lipid accumulation in 3T3-L1 cells. Administration of BEE to HFD-fed mice decreased body weight gain with the decreased serum cholesterol and triglyceride levels and reduced adipogenic gene expression, compared with vehicle-treated HFD-fed mice. The results are basically sound. However, there are concerns that should be addressed.
- Although in vitro study using 3T3-L1 cells was carried out, the results are not found in Abstract. The description about these results should be included in Abstract.
- Yes, added more information on the result of 3T3-L1 cell test. “The formation of lipid droplets in the 3T3-L1 treated with BEE (500 and 750 µg/ml) was signifi-cantly blocked (p < 0.05 and p < 0.01, respectively) “.
- The pictures of the cells in Figures 1 and 5 are not clear. They should be exchanged to clearer ones.
- It has been replaced with high-resolution pictures.
- The picture of representative animals after treatment had better been shown.
Unfortunately, we do not have full pictures of animals or pictures taken during dissection. We have only micrographs for tissue observation. please understand.
- Why was body weight gain in the BEE group decreased (Figure 2b), although food intake was increased after 24 days, compared with HFD group (Figure 2a)? The authors should explain this reason.
Yes, we added “Interestingly, after the 24th day, in the case of the BEE-administered group, food intake increased, but compared to the HFD control group, the body weight decreased. These results are considered to be related to the postprandial blood glucose rise inhibitory effect of the Arginyl-fructose (main key compound in BEE) reported in the previous study [13]. In addition to the regulation effect of BEE on adipogenesis/lipogenesis, it indicates that the effect of inhibiting absorption of ingested carbohydrates into the small intestine also may acted, giving a synergistic effect” in the section of results and discussion.
- Only 54 days-administration is not “long-term”. 54 day-treatment is quite common in these studies. The title should be revised.
- Yes, we all totally agree with your opinion, it has been removed.
- Table 2 and Figure 3 are redundant. Either one should be removed.
- It has been removed.
- How many animals were used in each study? They should be indicated in the legends of each figure and Table.
- 10 of each group, it was mentioned in the section of material and method “In this study, ten C57BL/6 mice were used for each group. The experimental protocols were approved by the Institutional Animal Care and Use Committee (IACUC) of the Hannam University (Approval number: HNU2020-0014)”.
Now, it has been mentioned in the legends of each figure and table.
- The expression of FAS was lowered in BEE group. However, although the authors stated that FAS is a key factor in lipogenesis, only one gene (FAS) is not enough. To mention the effect to lipogenesis, at least SREBP-1c, a key transcription factor in lipogenesis and other lipogenic gene such as SCD should be analyzed.
Yes, we added the result of the expression level of SREBP-1c in figure and
materials/methods, “SREBP-1c participated in adipocyte differentiation and adipogenesis is a major regulator of lipid homeostasis transcription [19]. Our observations demonstrated that PPARγ (p < 0.05) and C/EBPα (p < 0.001) mRNA level was significantly decreased with administration of BEE (0.3 g/kg-body weight) (Figure 4b and d). Additionally, we observed that BEE supplementation markedly decreased the expression levels of key target lipogenic genes, fatty acid synthase (FAS) (p < 0.001), lipoprotein lipase (LPL) (p < 0.01), and fatty acid binding protein 4 (FABP4) (p < 0.001), and sterol regulatory element-binding protein 1c (SREBP-1c) (p < 0.01) (Figure 4a, c, e, and f), an observation that correlates with the reduced PPARγ and C/EBPα mRNA expression”.

Reviewer 2 Report
There are some major issues with the paper, although the work may be of interest to some:
- Abstract is unclear and needs to be rewritten
- Results - repetitions observed, figures and tables lack clarity and presented in an uninteresting way, can be drastically improved
- Most important is that the discussion lacks critical discussion
- Lack of relevant references

Author Response
Thank you for your invaluable review, and advice. Corrections and supplements have been completed throughout the entire manuscript for all pointed out points.
There are some major issues with the paper, although the work may be of interest to some:
- Abstract is unclear and needs to be rewritten
- Yes, Abstract has been rewritten.
- Results - repetitions observed, figures and tables lack clarity and presented in an uninteresting way, can be drastically improved.
- Thank you for your suggestion, Yes, we all totally agree with your opinion, it has been improved through entire manuscript.
- Most important is that the discussion lacks critical discussion.
- Yes, based on more specific and objective research results, detailed explanations were added again. In addition, 9 references related to the description were added.
- Lack of relevant references
- More relevant and recently published references were replaced, and 9 new references related to explanations were added.

Round 2
Reviewer 1 Report
The manuscript was improved. I have no further comment.
Reviewer 2 Report
Accept